# The Effectiveness of Two Interventions for Improving Knowledge of Emergency Preparedness Amongst Enrollees of the World Trade Center Health Registry: A Randomized Controlled Trial

**DOI:** 10.3390/ijerph22071082

**Published:** 2025-07-07

**Authors:** Howard E. Alper, Lisa M. Gargano, Meghan K. Hamwey, Lydia F. Leon, Liza Friedman

**Affiliations:** 1New York City Department of Health and Mental Hygiene, World Trade Center Health Registry, 30–30 47th Avenue, Room 414, Long Island City, New York, NY 11101, USA; mhamwey@health.nyc.gov (M.K.H.); lleon1@health.nyc.gov (L.F.L.); liza.friedman@gmail.com (L.F.); 2Rhode Island Department of Health, 3 Capitol Hill, Providence, RI 02908, USA; lisa.gargano@health.ri.gov

**Keywords:** 9/11 Disaster, emergency preparedness, randomized controlled trials

## Abstract

Natural and man-made disasters are occurring more frequently, making household emergency preparedness essential for an effective response. Enrollees of the World Trade Center Health Registry have been found to be less prepared than the US national average despite their prior disaster exposure. The purpose of this study was to evaluate and compare the effectiveness of two interventions—a mailed brochure and a structured phone call—for increasing emergency preparedness knowledge among this population. We conducted a two-arm parallel group trial between February 2019 and August 2020. Participants were Registry enrollees who completed the Wave 4 Registry (2015–2016) survey, whose primary language was English or Spanish, who lived in New York City, and who did not report being a rescue and recovery worker affiliated with FDNY or NYPD. Enrollees were randomized to receive either a brochure by mail summarizing the components of emergency preparedness or a 15 min phone call describing the same. The primary outcome measure was the number of “yes” responses to the ten-item CDC CASPER emergency preparedness questionnaire, measured at baseline and post-intervention. Enrollees were sequentially alternatively assigned to either the brochure or phone call groups. In total, 705 enrollees were assigned to the brochure (*n* = 353) or phone call (*n* = 352) groups, and a total of 702 enrollees were analyzed. The Incident Rate Ratio (IRR) for the effect of time was 1.17 (95% CI = (1.14, 1.20)) and for intervention was 1.00 (95% CI = (0.95, 1.05)) Both the brochure and phone call interventions improved knowledge of emergency preparedness from baseline to post-intervention assessment, and to the same extent.

## 1. Introduction

Over the past several decades natural and human disasters have increased substantially, bringing mass devastation to the people and places impacted [1]. New York City has experienced multiple disasters including extreme weather events (e.g., Hurricane Sandy 2012), infectious disease outbreaks (e.g., Ebola, 2014; Measles, 2019; COVID-19, 2020), and terrorist attacks and attempts (e.g., WTC Truck Bombing 1993; 9/11 WTC Attacks, 2001). Following a disaster, help from local officials and first responders may be delayed for hours or days. Thus, risks for developing health problems from exposure to the elements, food and water shortages, damaged sanitation facilities, and impaired communication lines can lead to excess morbidity [2]. In the aftermath of Hurricane Sandy, for example, many experienced food and water shortages and were left without power or heat [3]. Thus, it seemed that a large proportion of those impacted by Sandy did not have an adequate stockpile of supplies or a clear emergency evacuation plan to mitigate the adverse effects of the hurricane [4].

Given the resulting personal and public hazards following a disaster, it is critical that households and communities are adequately prepared for emergencies to help mitigate these effects. Research has demonstrated that only 40–50% of households in the United States are disaster prepared [5], such that they have, readily available, an essentials kit, as well as an established disaster plan [2]. There is thus a pressing need for effective public health interventions to increase household preparedness by addressing key barriers for and motivators to increase preparedness.

While various interventions—ranging from brochures and leaflets to digital tools and expert-led sessions—have aimed to improve preparedness, most studies have focused on general populations and have not directly assessed individuals with prior disaster exposure. Notably, enrollees in the World Trade Center Health Registry (WTCHR), despite their direct experience with 9/11 and other disasters, report significantly lower preparedness levels than the national average—only 18.8% compared to 25% [6]. This paradox suggests a unique psychosocial profile within this group, possibly influenced by “optimistic bias,” where prior survival diminishes perceived vulnerability to future events [7].

Mixed degrees of success have been found among studies which employed pamphlets and leaflets in their intervention methodology. For instance, Barik and colleagues compared the use of pamphlets to posters and found that pamphlets were particularly effective for adult populations, as opposed to child/adolescent and elderly populations [8]. Similarly, Beaujean and colleagues compared leaflets and movies for increasing subject’s knowledge of Lyme disease in the Netherlands, demonstrating that both methods were more effective than a control group in increasing knowledge [9].

Recent research has employed more complex and resource-intensive interventions. For example James and colleagues (2020) integrated mental health and emergency preparedness training through expert-led discussions, which resulted in increased preparedness and reduced symptoms of mental health disorders, such as anxiety and PTSD [10]. Noor used a health belief-based intervention to improve knowledge and skills in emergency preparedness, showing significant gains compared to a control group [11]. Similarly, Adiyaman and colleagues found that earthquake preparedness training significantly improved psychological resilience and preparedness among mothers of children with physical disabilities [12].

These studies illustrate the diversity in approaches to improving preparedness and highlight the potential of more intensive interventions. However, no study has compared the use of pamphlets to other delivery modalities in a disaster-exposed population such as the World Trade Center Health Registry. Given the linguistic diversity of New York City, it was also critical to include Spanish-speaking individuals to ensure equitable access to emergency preparedness information; language-appropriate outreach is essential in a city where nearly one in four residents speaks Spanish at home, particularly in high-stakes disaster contexts where timely understanding can save lives. As such, given the gaps and lack of evidence on how best to improve preparedness in this population, we conducted a randomized controlled trial to compare the effectiveness of two practical interventions: a mailed brochure and a phone-based preparedness discussion. These particular approaches were utilized for their accessibility, scalability, and real-world applicability in large-population public health settings. The purpose of the study was to determine whether one approach was superior to the other in improving knowledge of emergency preparedness. This research seeks to inform future public health strategies for improving emergency readiness among disaster-exposed populations.

## 2. Methods

### 2.1. Study Design

Since initially we did not know which method was best for increasing awareness and knowledge of emergency preparedness, we wanted to compare two practical approaches to determine this. The best way to accomplish this methodologically was to compare the two emergency preparedness approaches using groups that demonstrated shared characteristics, which in our study, is having been exposed to the 9/11 Disaster. By ensuring a similar shared characteristic, we were able to then assess how the design approach employed may improve knowledge of emergency preparedness. The best method for achieving such a comparison is the randomized controlled trial (RCT), in which subjects are randomly assigned to treatment groups, ensuring the two groups are statistically the same. Therefore, a two-arm parallel group randomized controlled trial was designed to compare the effectiveness of a phone-based household emergency preparedness intervention with a mailed informational brochure on household emergency preparedness amongst a sample of World Trade Center Health Registry enrollees residing within New York City. This RCT was registered with www.clinicaltrials.gov; Identifier: NCT06737510.

### 2.2. Participants

The World Trade Center Health Registry comprises a longitudinal panel cohort of 71,423 enrollees who were exposed to the World Trade Center attacks in New York City on 11 September 2001. The Registry’s methods have been further described in prior publications [13]. The WTCHR protocol was approved by the Centers for Disease Control and Prevention (CDC) and NYC Department of Health and Mental Hygiene (DOHMH) institutional review boards (IRB # 02-058) [13]. All enrollees consented to be in the Registry. The protocol for the present study was approved by the DOHMH IRB on 21 November 2018. Enrollee informed consent for the present study was obtained during the initial outreach phone calls which were conducted between 2003 and 2004. All enrollees in the present study consented to participate. The subjects for this study were randomly selected from the Registry’s Wave 4 survey, subject to the following constraints: English and Spanish-speaking Registry enrollees who were older than 18 years on 9/11, resided in New York City, completed the Registry’s Wave 4 survey, and who were not 9/11 rescue/recovery workers for the New York Police Department (NYPD) or Fire Department of New York City (FDNY). Results from the present study were expected to apply to the entire Registry population. This trial is registered with clinicaltrials.gov (# NCT06737510).

### 2.3. Intervention

A brochure intervention to enhance household emergency preparedness was developed to include the following topics: (1) a brief statement regarding why emergency preparedness is important; (2) how to make a household disaster plan (or family communication and evacuation plan); (3) how to assemble an “emergency supply kit” of stockpiled supplies; (4) instructions on how to pack a “go bag” and what it should contain; and (5) an emergency reference card for all household members, with essential information (name, meeting place nearby or outside of neighborhood, contact numbers, evacuation location, doctors’ names and numbers). The brochure, entitled “Ready New York: Preparing for Emergencies,” was a pocket guide, available from NYC Emergency Management in several languages (https://www.nyc.gov/assets/em/downloads/pdf/rny_pocket_english.pdf, accessed on 12 January 2025). In addition to the information mentioned above, it included several resources: disaster contact numbers for emergency and non-emergency calls: 911 or 311, NYC Emergency Management, as well as a website for additional information on preparedness: NYC.gov/hazards, with information specific to New York City hurricane evacuation zones, and how to receive free emergency notifications.

The informational brochure intervention followed the format and topics above and was mailed to participants in the appropriate languages during the intervention period. The phone-based intervention consisted of a 15–20 min talk session completed over the phone following the format and topics noted for the brochure group. The phone intervention mentioned and offered the link to the NYC.gov/hazards website. Features of the two interventions are summarized in Table 1.

### 2.4. Procedures

This study consisted of a baseline survey, then the application two months later of one of two possible interventions designed to increase emergency preparedness over time, followed by a follow-up survey two months after that. The purpose of this study was to compare the effectiveness of the two interventions, among a sample of English- and Spanish-speaking Registry enrollees who resided in NYC and completed the Registry Wave survey (W4) in 2015–2016. Baseline information was collected from all participants over the phone and used a modified CASPER preparedness template, a reliable and valid epidemiological survey designed to gather individual household information about a community to respond to needs and make informed policy and programmatic decisions [14]. The baseline survey was designed to assess three domains: (1) basic household information; (2) household emergency and evacuation plans; and (3) household communication methods. The follow-up survey, also conducted by phone, was administered after the intervention to evaluate changes in enrollees’ level of emergency preparedness.

Between February 2019 and March 2020, eligible enrollees were identified and contacted in batches of 100 enrollees. Introductory (“Lead”) letters were mailed to a total of 2000 enrollees who completed the Wave 4 survey, including 400 Spanish- speaking enrollees, and excluding any enrollee who reported being a rescue and recovery worker affiliated with FDNY or NYPD due to their increased emergency response training. Approximately two weeks after the initial mailing, Registry staff members conducted the initial outreach phone call, whereby verbal consent to participate was obtained, and the baseline data was subsequently collected.

### 2.5. Randomization

Lead letters were sent out to participants in batches of 100. Enrollees who consented were alternately assigned to the brochure or phone call groups to ensure balance between the two intervention groups. Neither the researchers nor participants were blinded to intervention assignment.

A total of 1800 enrollees were assessed for eligibility and 1787 received a recruitment telephone call. Of those receiving a call, 1082 enrollees could not be contacted; 705 enrollees consented to participate in the study (Figure 1). Due to a programming error in the assignment of enrollees to interventions, the brochure and phone call groups developed a temporary imbalance. This was corrected by assigning 68 consecutively consented enrollees to the brochure group.

### 2.6. Outcomes

The primary outcome was emergency preparedness, assessed using a 10-item version of the CDC’s Community Assessment for Public Health Emergency Response (CASPER) questionnaire [14]. Each question measured a binary (yes/no) indicator of specific preparedness behaviors or resources. The questions covered two main domains: (1) communication planning (e.g., having a family meeting place, copies of documents, and an emergency communication plan), and (2) preparedness supplies (e.g., emergency food, water, medication, and go-bag items). The total score, ranging from 0 to 10, represented the number of preparedness actions reported. A higher score indicated greater household preparedness. Respondents were considered well-prepared if they scored 9 or 10 out of 10.

#### 2.6.1. Baseline Preparedness

Baseline emergency preparedness was captured from February 2019 to March 2020, using an adapted version of the CASPER instrument [14]. A total of ten questions were asked, rated on a dichotomous scale. Sample items include: Does your household have any of the following emergency plans? Emergency Communication plan such as a list of numbers and designated out of town contact. Designated meeting place immediately outside your home or close by in your neighborhood. Copies of important documents in a safe location (waterproof container). Each response was assigned a value of one for a “yes” and zero for a “no”. The total emergency preparedness score was the sum of the ten responses, ranging from 0 (not prepared) to 10 (completely prepared). Respondents were considered prepared in this study if they scored a 9 or10. These questions included addressing whether participants had disaster supplies and emergency communication plans. Other components of the primary outcome included the total score over the five communication questions, the total score over the five emergency kit questions, and each of the ten individual CASPER items. The ten questions of the CASPER instrument are listed in the Appendix A.

#### 2.6.2. Follow-Up Preparedness Variables

The CASPER instrument was also used to measure emergency preparedness in the post-intervention follow-up evaluation, two months after the intervention and four months after the baseline survey.

### 2.7. Wave Data Variables

The Registry conducted surveys (called “waves”) of its enrollees every four to five years. The initial survey (Wave 1) took place from 2003 to 2004, with subsequent surveys taking place during 2006–7 (Wave 2), 2011–12 (Wave 3), and 2015–16 (Wave 4). Wave 1 fielded questions about basic demographics, exposure to the disaster, and pre-9/11 and post-9/11 chronic disease. Wave 2 contained further detailed questions relating to enrollees’ 9/11 exposure, and Waves 2–4 asked about the subsequent development of chronic physical and mental disease.

#### 2.7.1. SocioDemographics Variables

Demographic characteristics included were age at the Wave 1 survey, sex collected at Wave 1, race/ethnicity from the Wave 1 survey, education from the Wave 1 survey, income at Wave 3 or Wave 4, and recruitment source (List: Enrollee located from list of lower Manhattan building inhabitants; Self: Enrollee responded to media campaign).

#### 2.7.2. 9/11 Exposure

Participant exposure at 9/11 was measured using a derived variable comprising eleven items developed from the Wave 1 and wave 2 surveys. Sample items included the number of horrific events witnessed on 9/11, dust cloud exposure intensity, and general measure of number of 9/11 exposures (e.g., being in the north or south WTC Tower or other fully collapsed building on 9/11; moving out of home for at least 24 h between 9/11 and 9/18). This measure has been used and validated in several published studies [13,15,16]. For the present study the 9/11 exposure was collapsed to four categories: low (0–1), medium (2), high (3), and very high (4–11).

#### 2.7.3. Exposure to Hurricane Sandy

Disaster exposure during Hurricane Sandy was measured using seven categories captured using dichotomous (Yes/No) responses. Enrollees could select all the options that applied. These categories included: exposure to sewage, debris, dirty or contaminated flood water, visible mold, exhaust fumes from generators, diesel fuel or heating oil leaks or spills, or no exposure. This measure has been validated in other published works [6]. This Sandy exposure was collapsed into a dichotomous variable.

### 2.8. Analytic Strategy

To evaluate baseline emergency preparedness for the two interventions, means and frequency statistics were calculated for the baseline sample to evaluate the association of sociodemographic variables from the Wave 1, 2, 3, and 4 surveys with the intervention (brochure vs. phone call). A χ^2^ test was employed for categorical variables, where a *p*-value > 0.05 meant there was no association between a particular sociodemographic variable and the intervention. A *t*-test was employed to compare the association of continuous variables with the intervention. We also calculated frequencies for the emergency preparedness items by the intervention.

The goal of the main analysis was to determine if there was a statistically and substantially significant advantage of one intervention over the other in improving knowledge of emergency preparedness. Since there were repeated measures per enrollee (e.g., the baseline and post-intervention preparedness evaluations), we tested our main hypothesis, stated above, using Generalized Estimating Equations (GEE) following the intention-to-treat (ITT) principle. We ran GEE regressions separately for the total CASPER preparedness score, the total score for preparing an emergency kit, the total score for preparing communication plans, and each of the ten individual items. GEE regressions on the three total CASPER scale sums used a negative binomial distribution with a log ink, while GEE regression on individual CASPER items used a Poisson distribution with a log link (the latter because the prevalence of many items was greater than 10 percent) and robust standard error estimates. Multiple comparison corrections were applied to GEE regressions for individual CASPER items. Since some enrollees were missing the primary outcomes at the post-intervention assessment, we employed Multiple Imputation under the Missing at Random (MAR) assumption to fill in the missing outcome data for the ITT analysis, using the SAS-compatible program IVEware version 0.3 (University of Michigan, Ann Arbor, MI, United States). The Per-Protocol (PP) approach was also employed. It included only those enrollees who followed the protocol. Since no enrollees switched treatment arms during the trial, the Per Protocol approach was equivalent to a “complete cases” analysis. Since this study overlapped with the start of the COVID pandemic, we performed as a sensitivity analysis of the Per-Protocol analysis separately for enrollees who completed the post-assessment before vs. after March 2020.

## 3. Results

### 3.1. Demographics

Comparison of sociodemographic (age, race, gender, education) and other variables (rescue/recovery worker status, injured on 9/11, 9/11 exposure scale, recruitment source, social support at wave 4, one or more chronic physical conditions at wave 4, exposure to Hurricane Sandy) (Table 2) between the brochure and phone groups found no substantial or statistically significant differences at baseline, except for social support at wave 4 and recruitment source.

### 3.2. Test of Main Hypothesis

Table 3 presents the results for the degree of emergency preparedness at baseline for the brochure and phone call groups. The two interventions exhibited a comparable baseline degree of preparedness for the total CASPER scale, the communication and kit sub-scales, and the ten individual items. However, the degree of preparedness was not uniformly high. For example, the total CASPER scores for both interventions were only about 5.5/10, indicating only moderate levels of preparedness. The communication scale, with average values of 2.3, and the kit scale, with values around 3.3, both also correspond to only modest levels of preparedness. The probabilities for individual preparedness items were often quite low, though some were very high (e.g., food for three days, medication for seven days).

Table 4 presents results comparing the brochure and phone interventions. For the Per Protocol (PP) analysis, the total CASPER score sum increased by 17% from baseline to post-assessment (IRR = 1.17, 95% CI = (1.14, 1.20)), while the communication sub-scale sum increased by 23% (IRR = 1.23 95% CI = (1.17, 1.28)) and the kit sub-scale-sum increased by 13% (IRR = 1.13 95% CI = (1.09, 1.18)) over the same period. However, there was no substantial or statistically significant difference between the effects of the two interventions over time for any of these scales. These results were not substantially or statistically different when conducted separately for enrollees who entered the study before vs. after March 2020 (the COVID pandemic; or when stratified by (separately) 9/11 exposure or exposure to Hurricane Sandy. Similar results were obtained for the Intention to Treat (ITT) analysis. For the individual CASPER scale items, Figure 2 demonstrates that items showed varied increases in the probability of an affirmative response from baseline to post-assessment, with some showing substantial increases (e.g., Meeting Place Outside RR = 1.70, 95% CI = (1.36, 2.12)). However, like the PP analysis, the two interventions showed quite similar effects over time.

## 4. Discussion

We compared the effects of a brochure mailed to one group of enrollees of the World Trade Center Health Registry with a phone call to another group, to determine if one method was more effective at increasing knowledge of emergency preparedness. We assessed changes using the total CASPER score, its communication and kit subscales, and the ten individual CASPER items.

We found that emergency preparedness increased over time from baseline to post-intervention, with overall preparedness rising by 17%. Some individual items improved by as much as 70%. However, we observed no statistically significant difference between the two intervention groups over time; brochures and phone calls were, on average, equally effective at improving preparedness.

The effect size achieved here, while statistically significant, was modest. The average preparedness score rose from approximately 5.5 to 6.5 out of 10—well below the threshold of 9 or 10 typically associated with being “well-prepared” in the event of a real emergency. This raises questions about the practical significance of the findings, suggesting that while the interventions were helpful, they may not have translated into comprehensive readiness.

Several factors may explain these modest gains. Psychological readiness—including prior trauma exposure and perceived threat—could have influenced participants’ motivation or ability to act. Educational background and health literacy may also have played a role, particularly for those who found the materials difficult to interpret or implement. Moreover, the interventions were delivered only once, limiting their potential to drive sustained behavior change.

The interventions employed here were low-intensity, and the absence of a no-intervention control group further limits interpretation. Nonetheless, our results align with earlier studies showing modest gains from minimal interventions [9,17] but contrast with findings from more intensive strategies. For example, an RCT with Latinos in Los Angeles found substantially greater improvements in preparedness among those participating in facilitated small group discussions (“pláticas”) compared to those receiving mailed materials [18]. Similarly, other studies employing face-to-face education or group-based training reported preparedness gains of 30–44% over controls [10,12].

Importantly, while the average effect here was similar across groups, future research should explore whether specific populations—such as older adults or those with limited literacy—might benefit more from phone-based or interactive formats. These individuals may face barriers in engaging with written materials and could benefit from more accessible, personalized communication. In a diverse urban setting like New York City, tailoring interventions to reach these subgroups could improve equity and overall effectiveness.

These findings highlight the limits of low-touch interventions and point to the need for more intensive, sustained, or community-embedded strategies. Future programs might incorporate multiple contact points, peer or community leader engagement, or culturally tailored content. Embedding preparedness messaging within trusted service systems—such as healthcare or social services—could also support longer-term behavior change.

This study had several limitations. First, randomization was briefly disrupted, causing a temporary imbalance between groups. This was corrected quickly, and baseline analysis confirmed balance on key covariates but this departure from strict randomization could have had residual effects on the analysis in that the two intervention groups may not have been strictly comparable on variables not included in the baseline analysis. Second, no information was collected on whether brochure recipients received or read the materials, limiting insight into engagement This means that we cannot know the “dose” of the brochure intervention and if it was equivalent to that for the phone call (i.e., outcome equivalence). Third, there was no true control group receiving no intervention; future studies should include such a group for clearer effect attribution. Fourth, although the study was conducted partly during the COVID-19 pandemic, analyses showed no significant differences between participants enrolled before and during the pandemic. Nonetheless, pandemic-related stressors may have influenced responses and priorities. Fifth, this study was limited to English- and Spanish-speaking subjects in an urban environment. While it is likely that this study’s protocol could be applied to speakers of other languages in an urban setting, this protocol would be more difficult to apply in dispersed rural areas.

Academically, this study contributes to emergency preparedness research by providing one of the few direct comparisons of two low-intensity intervention modalities in a disaster-exposed population, using validated CASPER metrics. The focus on World Trade Center Health Registry enrollees adds insight into preparedness behavior among individuals with shared trauma histories. The study also provides a replicable evaluation framework for future work in this area.

Practically, our findings suggest that while low-cost, scalable interventions like brochures and phone calls can lead to modest improvements, they may be insufficient for achieving real-world readiness. Public health agencies should consider combining such strategies with higher-impact approaches, particularly when targeting vulnerable or hard-to-reach populations. Even small gains, however, are meaningful when deployed at scale—especially in post-disaster contexts. This study underscores the importance of reaching populations with simple preparedness messaging, while also calling attention to the need for more targeted, equity-informed approaches to risk communication.

## 5. Conclusions

In conclusion, our study found that both a mailed brochure and a phone call equally increased emergency preparedness among World Trade Center Health Registry enrollees, with a 17% overall improvement and up to 70% increases in individual items. However, no significant differences between the two interventions were observed. These results align with previous research on low-impact interventions but differ from studies using more complex methods. The lack of a control group and the relatively low intensity of the interventions may explain the modest outcomes. Despite study limitations, including compromised randomization and the absence of post-assessment data for the brochure group, our findings suggest that simple interventions can improve preparedness to a modest degree, but more comprehensive strategies may be needed for greater impact.

## Figures and Tables

**Figure 1 ijerph-22-01082-f001:**
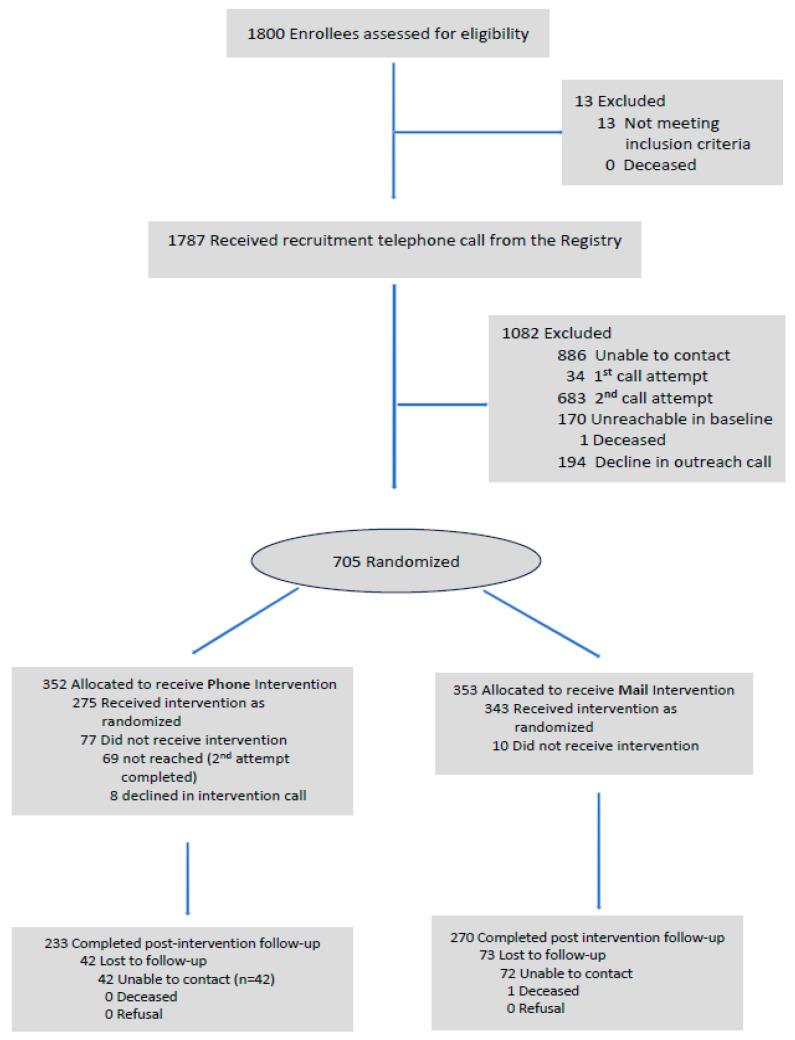
Consort diagram. Enrollee recruitment and study flow.

**Figure 2 ijerph-22-01082-f002:**
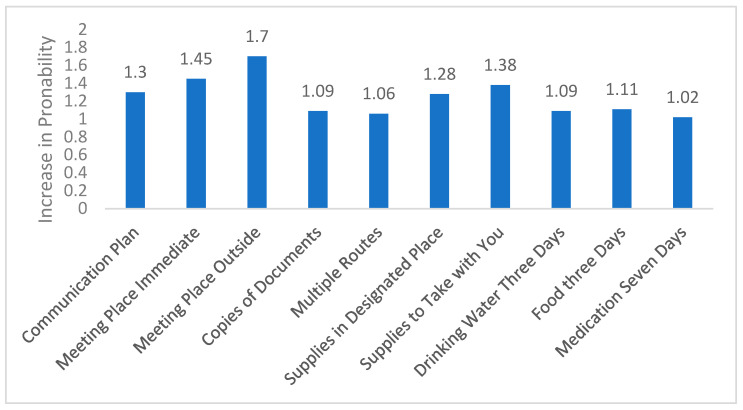
Change over time in probability of a positive response for individual CASPER items. Note: This figure depicts the ratio, from baseline to post assessment, of the probability of a positive response for each of the ten CASPER items assessed.

**Table 1 ijerph-22-01082-t001:** Features of the brochure and phone call interventions.

Feature	Brochure Intervention	Phone Call Intervention
Delivery Method	Mailed pocket brochure titled “Ready New York: Preparing for Emergencies”	12–20 min phone conversation
Languages Offered	English and Spanish	English and Spanish
Main Content Topics	Importance of preparedness, how to make a plan, kit contents, go-bag, emergency contacts	Same as brochure
Resources Provided	Contact numbers (911, 311), website links (NYC.gov/hazards)	Contact numbers (911, 311), website links (NYC.gov/hazards) mentioned
Personalization	None	Registry staff available to clarify content and answer questions
Follow-up Encouraged	Not Applicable	Participants encouraged to visit linked resources post-call

**Table 2 ijerph-22-01082-t002:** Baseline characteristics of the analytic sample by intervention.

Characteristic	Total		Brochure		Phone Call		*p*-Value
	*n*	%	*n*	%	*n*	%	
Total Sample	702	100.0	352	50.0	350	50.0	
Gender							0.82
Male	334	47.5	166	49.7	168	50.3	
Female	368	52.5	186	50.5	182	49.5	
Age at Wave 1							0.93
0–44	320	45.6	161	50.3	159	49.7	
45–above	382	54.4	191	50.0	191	50.0	
Race							0.48
Non-Hispanic white	412	58.7	202	49.0	210	51.0	
Other	290	41.3	150	51.7	140	48.3	
Education							0.38
High school	164	23.5	80	48.8	84	51.2	
Some college	159	22.8	78	49.1	81	50.9	
College	229	32.8	125	54.6	104	55.4	
Grad School	146	20.9	67	45.9	79	54.1	
Physical Health Conditions							0.48
No	94	86.7	44	46.8	50	53.2	
Yes	608	13.3	308	50.7	300	49.3	
Injury at Wave 1							0.29
0	594	85.2	292	49.1	302	50.9	
1	80	11.5	42	52.5	38	47.5	
2+	23	3.3	15	65.2	8	34.8	
Social Support at Wave 3							0.06
Low	356	52.9	192	53.9	164	46.1	
High	317	47.1	148	46.7	169	53.3	
RRW Status							0.97
Yes	493	70.2	105	50.2	104	49.8	
No	209	29.8	247	50.1	246	49.9	
Mental Health Conditions							0.18
0	550	78.4	271	49.3	279	50.7	
1	75	10.7	45	60.0	30	40.0	
2	77	11.0	36	46.8	41	53.2	
Hurricane Sandy							0.89
No	573	81.6	288	50.3	285	49.7	
Yes	129	18.4	64	49.6	65	50.4	
9/11 Exposure							0.70
Low	234	33.3	117	50.0	117	50.0	
Medium	147	20.9	68	46.3	79	53.7	
High	117	16.7	62	53.0	55	47.0	
Very High	204	29.1	105	51.5	99	48.5	
Recruitment Source							0.00
List	175	24.9	105	60.0	70	40.0	
Self	527	75.1	247	46. 7	280	53.3	

**Table 3 ijerph-22-01082-t003:** Baseline mean preparedness scores for the brochure and phone call intervention groups.

Quantity	Brochure	Phone Call
Total CASPER Score	5.64	5.55
Communications Score	2.29	2.22
Kit Score	3.34	3.33
Communication Plan	0.46	0.43
Meeting Place Immediate	0.26	0.29
Meeting Place Outside	0.25	0.20
Copies of Documents	0.66	0.68
Multiple Routes	0.66	0.63
Supplies in Designated Place	0.44	0.41
Supplies to Take with You	0.45	0.41
Drinking Water 3 Days	0.69	0.72
Food 3 Days	0.79	0.80
Medication 7 Days	0.97	0.98

**Table 4 ijerph-22-01082-t004:** GEE regressions for total CASPER scale, communication and kit sub-scales, and the ten individual CASPER items (brochure vs. phone call).

	Per Protocol		Intention to Treat	
Regression	Time (IRR/RR)	Intervention (IRR/RR)	Time (IRR/RR)	Intervention (IRR/RR)
Total CASPER Score	1.17 (1.14, 1.20)	1.01 (0.95, 1.07)	1.18 (1.15, 1.21)	1.00 (0.95, 1.05)
Communications Score	1.23 (1.17, 1.28)	1.03 (0.94, 1.13)	1.24 (1.19, 1.29)	1.00 (0.92, 1.08)
Kit Score	1.13 (1.09, 1.18)	0.99 (0.94, 1.05)	1.13 (1.10, 1.16)	1.01 (0.96, 1.05)
Communication Plan	1.30 (1.14, 1.47)	1.02 (0.84,1.25)	1.32 (1.20, 1.44)	1.01 (0.82, 1.18)
Meeting Place Immediate	1.45 (1.20, 1.75)	1.25 (0.93, 1.67)	1.48 (1.27, 1.68)	1.17 (0.89, 1.44)
Meeting Place Outside	1.70 (1.37, 2.12)	0.91 (0.67, 1.23)	1.71 (1.46, 2.00)	0.90 (0.63, 1.16)
Copies of Documents	1.09 (0.99, 1.20)	1.06 (0.92, 1.21)	1.09 (1.00, 1.18)	1.03 (0.91, 1.15)
Multiple Routes	1.06 (0.96, 1.17)	0.98 (0.85, 1.12)	1.08 (0.98, 1.18)	0.95 (0.82, 1.08)
Supplies in Designated Place	1.28 (1.11, 1.47)	0.99 (0.80, 1.22)	1.30 (1.15, 1.44)	0.97 (0.79, 1.16)
Supplies to Take with You	1.38 (1.20, 1.58)	0.93 (0.75, 1.16)	1.32 (1.18, 1.46)	1.00 (0.81, 1.18)
Drinking Water 3 Days	1.09 (1.01, 1.17)	1.00 (0.88, 1.14)	1.09 (1.01, 1.17)	1.02 (0.91, 1.13)
Food 3 Days	1.11 (1.04, 1.19)	1.03 (0.95, 1.12)	1.12 (1.05, 1.18)	1.02 (0.95, 1.10)
Medication 7 Days	1.02 (0.99, 1.04)	1.01 (0.98, 1.03)	1.02 (1.00, 1.04)	1.00 (0.98, 1.03)

## Data Availability

The data presented in this study are available on request from the corresponding author due to legal and privacy restrictions.

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
