# Peer review of "The Effectiveness of Two Interventions for Improving Knowledge of Emergency Preparedness Amongst Enrollees of the World Trade Center Health Registry: A Randomized Controlled Trial"

_ijerph, 2025, doi:10.3390/ijerph22071082_

Round 1
Reviewer 1 Report (Previous Reviewer 2)
Comments and Suggestions for Authors
The revisions and supplements to the first paper review were well done.
Author Response
This reviewer had no comments.
Reviewer 2 Report (New Reviewer)
Comments and Suggestions for Authors
- The use of a comparative intervention design is relatively uncommon in the field of emergency management, which lends methodological novelty to this study. This approach demonstrates innovation in evaluating preparedness interventions and could serve as a reference for future research designs in the field.
- By focusing on individuals who have experienced a major disaster, the study enhances its relevance to public health policy and disaster resilience planning. The sample selection increases the contextual validity and societal significance of the findings.
- The findings provide valuable implications for future emergency preparedness strategies. It is suggested that the authors consider discussing the applicability of the interventions to non-English-speaking populations.Such an addition would strengthen the study’s relevance to diverse, multilingual urban communities and promote inclusive preparedness planning.
- The two intervention methods—brochure and phone call—are both practical and scalable. However, it is recommended to clarify whether participants’ age distribution was taken into account, as age may influence the accessibility and effectiveness of the intervention. Age-related differences in information processing or communication preferences could affect outcomes.
- Although both interventions yielded comparable improvements in preparedness, further analysis could explore whether phone-based interventions are particularly effective for individuals with limited literacy or for older adults. This would add practical value to the study by identifying subgroups who may benefit from tailored intervention approaches.
- The average CASPER preparedness score in both groups was approximately 5.5 out of 10, indicating a moderate level of preparedness. The authors are encouraged to discuss potential factors contributing to this result, such as psychological readiness, educational background, or the influence of the COVID-19 pandemic on participant behavior. This context would help interpret the preparedness baseline and post-intervention outcomes more comprehensively.
- Considering the group size imbalance and the overlap of data collection with the COVID-19 pandemic, it is recommended to elaborate on how these external variables were managed or adjusted for in the analysis.
- The conclusion could be strengthened by more clearly articulating the study’s academic contributions and practical implications for emergency preparedness and risk communication. This would better demonstrate the study’s value to both scholars and practitioners.
- It is recommended that future research include a true control group with no intervention, to allow a more rigorous evaluation of intervention effectiveness. Such a design would help isolate the specific impact of each intervention and improve causal inference.
Author Response
|
Reviewer 2 |
Response |
Page(s) |
|
|
|
|
|
The use of a comparative intervention design is relatively uncommon in the field of emergency management, which lends methodological novelty to this study. This approach demonstrates innovation in evaluating preparedness interventions and could serve as a reference for future research designs in the field. .
|
We thank the reviewer for their thoughtful comment.
|
|
|
By focusing on individuals who have experienced a major disaster, the study enhances its relevance to public health policy and disaster resilience planning. The sample selection increases the contextual validity and societal significance of the findings.
|
Thank you for identifying this, we appreciate your comment.
|
|
|
The findings provide valuable implications for future emergency preparedness strategies. It is suggested that the authors consider discussing the applicability of the interventions to non-English-speaking populations. Such an addition would strengthen the study’s relevance to diverse, multilingual urban communities and promote inclusive preparedness planning.
|
We appreciate this comment and have added details to the introduction to better elucidate the importance of providing this type of information in languages other than English. (see page 2)
|
2 |
|
The two intervention methods—brochure and phone call—are both practical and scalable. However, it is recommended to clarify whether participants’ age distribution was taken into account, as age may influence the accessibility and effectiveness of the intervention. Age-related differences in information processing or communication preferences could affect outcomes.
|
As part of the research design, we opted to conduct the study using randomized methods, so age was not taken into consideration in doing this to best ensure true randomization. In the results and table 2 we also explain in more detail how we assessed age as baseline characteristic differences, finding that age was not differentially related to any baseline characteristic in the study.
|
8 |
|
Although both interventions yielded comparable improvements in preparedness, further analysis could explore whether phone-based interventions are particularly effective for individuals with limited literacy or for older adults. This would add practical value to the study by identifying subgroups who may benefit from tailored intervention approaches.
|
We appreciate the reviewer’s thoughtful identification of this need. We have added additional details to the discussion (see page 13) to better address the benefits of this method of these populations.
|
12-13 |
|
The average CASPER preparedness score in both groups was approximately 5.5 out of 10, indicating a moderate level of preparedness. The authors are encouraged to discuss potential factors contributing to this result, such as psychological readiness, educational background, or the influence of the COVID-19 pandemic on participant behavior. This context would help interpret the preparedness baseline and post-intervention outcomes more comprehensively.
|
Thank you for pointing this out. We have added some more details around this in the discussion section of the manuscript (see page 13). We would like to point out as well that we did assess the impact of COVID-19 in terms of differences among those who enrolled prior to and during COVID, finding little to no differences between the two groups of individuals. We have highlighted that point and made it more readily identifiable as well.
|
12-13 |
|
Considering the group size imbalance and the overlap of data collection with the COVID-19 pandemic, it is recommended to elaborate on how these external variables were managed or adjusted for in the analysis.
|
We state in the results (page 8) and discussion (page 13) that the temporary imbalance in group size was quickly corrected and the two groups were balanced on the majority of baseline variables.
We also performed the main analysis stratified by whether enrollees entered the study before or during the COVID pandemic and found no statistical or substantial difference by this variable.
Wanting to look at this a bit further, we have re-analyzed the data to determine group stratification differences. Upon review the group that received the brochure were slightly younger, and the group who received phone only were slightly older. However, when we looked at statistical differences between the groups, across all factors, we found no statistical differences.
|
8, 13 |
|
The conclusion could be strengthened by more clearly articulating the study’s academic contributions and practical implications for emergency preparedness and risk communication. This would better demonstrate the study’s value to both scholars and practitioners.
|
Thank you for this identification. We have added additional details to the manuscript on pages 12 and 13 to better incorporate the implications for academics, as well as public health more broadly.
|
13 |
|
It is recommended that future research include a true control group with no intervention, to allow a more rigorous evaluation of intervention effectiveness. Such a design would help isolate the specific impact of each intervention and improve causal inference.
|
Thank you for your helpful comment. We have added this point to the discussion in the limitations section (see page 13) to more clearly articulate that a true comparison group of no treatment received is needed.
|
13 |
Reviewer 3 Report (New Reviewer)
Comments and Suggestions for Authors
The authors conducted a two-arm randomized controlled trial to compare the effectiveness of two interventions—a mailed brochure versus a structured phone call—in improving household emergency preparedness among World Trade Center Health Registry (WTCHR) enrollees. The primary outcome was the change in preparedness based on the 10-item CDC CASPER questionnaire. The results showed a increase in preparedness across time for both groups, but no differential effect between interventions. This manuscript reports a methodologically sound and policy-relevant randomized controlled trial that addresses a highly pertinent public health concern—household emergency preparedness among disaster-exposed populations. However, several design limitations constrain the broader interpretability and practical impact of the results.
Comment 1: The manuscript would benefit from a clearer articulation of the study’s primary hypothesis and corresponding statistical framework. The authors state: “The main hypothesis was that the two interventions would increase emergency preparedness but would differ in the size of their effects.” However, it remains unclear whether the objective was to test a hypothesis of superiority, equivalence, or non-inferiority between the two interventions. Although a two-arm randomized controlled trial is methodologically appropriate for comparing interventions, the absence of a prespecified margin of difference and a corresponding analytical plan limits interpretability. Moreover, the Discussion section concludes that “no difference” was observed, suggesting that a superiority design may have been assumed—though not formally declared.
I recommend the authors explicitly state their hypothesis type in the Introduction and define the corresponding statistical interpretation framework (e.g., specifying whether the goal was to detect a statistically significant advantage of one intervention over the other, or to demonstrate equivalent effectiveness). This clarification will strengthen the scientific rigor and transparency of the trial’s objectives and align the results with their appropriate interpretive context.
Comment 2: Given the heterogeneity of exposure (e.g., 9/11 severity, Hurricane Sandy), a subgroup analysis might provide insights into differential responsiveness.
Comment 3: The authors should more explicitly define the boundaries of generalizability. The study population is primarily urban, English- or Spanish-speaking, older adults with prior disaster exposure.
Comment 4: A programming error caused an imbalance in participant assignment between intervention arms. Although the issue was later corrected manually, this departure from strict randomization raises concerns about internal validity. I recommend emphasizing this point more clearly in the Limitations section.
Comment 5: The study did not assess whether participants in the brochure group actually received, read, or engaged with the material. In contrast, the phone intervention ensured participant interaction. This limits the ability to interpret outcome equivalence, as the 'dose' of intervention may have differed significantly between groups.
Comment 6: Both interventions were “light-touch.” The modest increases in preparedness (17% overall) may reflect the low intensity of both approaches. The authors acknowledge this, but more discussion on how the intervention might be enhanced (e.g., repetition, multimedia) would be beneficial.
Comment 7: Despite statistical significance, practical significance is debatable. With mean CASPER scores moving from ~5.5 to ~6.5, the effect might be insufficient to translate into meaningful preparedness in a real emergency. Specifically, Table 3 shows that baseline preparedness levels, as measured by the total CASPER score, were approximately 5.5 out of 10 for both intervention groups, and only modestly improved post-intervention. Table 4 further indicates that the overall preparedness score increased by 17% (IRR = 1.17), which translates to a final average of roughly 6.5—still well below the threshold for "well-prepared" status, defined as a score of 9 or 10. In the Discussion, the authors note that “individual items increased by as much as 70%,” yet they also acknowledge that “we did not find any difference in the effect of the two interventions,” and compare their modest findings to more impactful results from recent high-intensity intervention studies. These observations confirm that, although statistically significant, the behavioral impact of the interventions was limited in magnitude and potentially insufficient to produce practical readiness in the event of a real disaster.
The authors are encouraged to contextualize the modest improvement in preparedness by discussing its practical significance and suggesting how future interventions might achieve stronger and more sustained behavioral impact.
Author Response
|
Reviewer 3 |
Response |
Page(s) |
|
|
|
|
|
The manuscript would benefit from a clearer articulation of the study’s primary hypothesis and corresponding statistical framework. The authors state: “The main hypothesis was that the two interventions would increase emergency preparedness but would differ in the size of their effects.” However, it remains unclear whether the objective was to test a hypothesis of superiority, equivalence, or non-inferiority between the two interventions. Although a two-arm randomized controlled trial is methodologically appropriate for comparing interventions, the absence of a prespecified margin of difference and a corresponding analytical plan limits interpretability. Moreover, the Discussion section concludes that “no difference” was observed, suggesting that a superiority design may have been assumed—though not formally declared.
I recommend the authors explicitly state their hypothesis type in the Introduction and define the corresponding statistical interpretation framework (e.g., specifying whether the goal was to detect a statistically significant advantage of one intervention over the other, or to demonstrate equivalent effectiveness). This clarification will strengthen the scientific rigor and transparency of the trial’s objectives and align the results with their appropriate interpretive context.
|
The reviewer raises an important point regarding the study design. We have taken their suggestion and rewritten the description of the main hypothesis at the end of the introduction and in the analytic methods section to highlight the hypothesis type of superiority with a prespecified margin of zero. |
2, 8 |
|
Given the heterogeneity of exposure (e.g., 9/11 severity, Hurricane Sandy), a subgroup analysis might provide insights into differential responsiveness. |
We welcome the reviewer’s suggestion. We have performed the main analysis stratified by, separately, the 9/11 exposure and Hurricane Sandy variables. We found that the results did not differ by level of 9/11 exposure or exposure to Hurricane Sandy.
|
10 |
|
The authors should more explicitly define the boundaries of generalizability. The study population is primarily urban, English- or Spanish-speaking, older adults with prior disaster exposure. |
To address this comment about generalizability boundaries, we have added a discussion of the applicability of the study interventions to different regions and languages in the discussion section. |
12-13 |
|
A programming error caused an imbalance in participant assignment between intervention arms. Although the issue was later corrected manually, this departure from strict randomization raises concerns about internal validity. I recommend emphasizing this point more clearly in the Limitations section.
|
Thank you for pointing out this concern, To address this more explicitly we have added discussion of the potential residual effects of the programming error on the balance between the intervention groups, in the discussion section. |
12-13 |
|
The study did not assess whether participants in the brochure group actually received, read, or engaged with the material. In contrast, the phone intervention ensured participant interaction. This limits the ability to interpret outcome equivalence, as the 'dose' of intervention may have differed significantly between groups. |
We appreciate the reviewer’s comment on determining true uptake of the information received in the brochure intervention by participants. We agree that this is an inherent challenge and caveat of the study and have added details to the discussion to better address this concern and make it clearer for readers. |
12-13 |
|
Both interventions were “light-touch.” The modest increases in preparedness (17% overall) may reflect the low intensity of both approaches. The authors acknowledge this, but more discussion on how the intervention might be enhanced (e.g., repetition, multimedia) would be beneficial. |
In the discussion section we have added material suggesting future enhancements to the approaches used in the present study. |
12-13 |
|
Despite statistical significance, practical significance is debatable. With mean CASPER scores moving from ~5.5 to ~6.5, the effect might be insufficient to translate into meaningful preparedness in a real emergency. Specifically, Table 3 shows that baseline preparedness levels, as measured by the total CASPER score, were approximately 5.5 out of 10 for both intervention groups, and only modestly improved post-intervention. Table 4 further indicates that the overall preparedness score increased by 17% (IRR = 1.17), which translates to a final average of roughly 6.5—still well below the threshold for "well-prepared" status, defined as a score of 9 or 10. In the Discussion, the authors note that “individual items increased by as much as 70%,” yet they also acknowledge that “we did not find any difference in the effect of the two interventions,” and compare their modest findings to more impactful results from recent high-intensity intervention studies. These observations confirm that, although statistically significant, the behavioral impact of the interventions was limited in magnitude and potentially insufficient to produce practical readiness in the event of a real disaster.
The authors are encouraged to contextualize the modest improvement in preparedness by discussing its practical significance and suggesting how future interventions might achieve stronger and more sustained behavioral impact.
|
We have added material to the discussion section to evaluate the practical significance of our study findings. We have also discussed approaches that would produce more sustained and stronger impact on knowledge of emergency preparedness. We have further added details to inform the practical use of this information and preparedness research to this same section at the request of Reviewer 2. We appreciate the reviewers identifying this need and have attempted to synthesize both comments into one cohesive section. |
12-13 |
This manuscript is a resubmission of an earlier submission. The following is a list of the peer review reports and author responses from that submission.
Round 1
Reviewer 1 Report
Comments and Suggestions for Authors
The author has made some corrections, but the location is not visible because the response on the page number does not match between the manuscript and the revision page. For example, there are no page numbers 33-37 in the manuscript! The author must verify that the manuscript pages align with the pages listed in the Response to Reviewers letter.
Author Response
|
Reviewer 1 |
Response |
Page(s) |
|
Background:** This section should provide a concise description of the research context, emphasizing the importance of the topic and the need for further investigation in this area.
|
We have restructured the introduction to directly emphasize the reviewer’s points concerning context, topic importance, and the need for further research. |
1- Top of 3 |
|
Include relevant keywords associated with the research topic to enhance search visibility and increase citations of the article.
|
We have expanded the keyword section. |
1 |
|
Ensure that the reference format adheres to the journal's guidelines. |
We have corrected the format of the relevant references. |
13-15 |
|
Clearly state the purpose of the study in the introduction.
|
We have stated the study purpose in the final paragraph of the introduction. |
3 |
|
Participants: There are inconsistencies in font sizes within this section; please standardize the font and size throughout. |
We have corrected the font sizes in this paragraph. |
3 (section 2.2) |
|
The phrase "Enrollee informed consent for the present study was obtained during the initial outreach phone call" is ambiguous. Did all participants agree to the informed consent presented? What was the protocol for those who declined? |
We have expanded the discussion of the informed consent process. |
3 |
|
Wave Data Variables: Define what the term "Wave" refers to in this context. Specify the data collection period and outline the differences between Wave 1, Wave 2, Wave 3, and Wave 4.
|
We have supplied our definition of “Wave” and described the data collection process in further detail. We also provided summaries of the information gathered at each wave. |
7 (section 2.7) |
|
Table 1: Clarify the meaning of the p-value and indicate which statistical test was employed |
We have clarified which statistical tests we used to summarize the descriptive results, and the purpose of the description. |
8 |
|
Table 1: Explain the terms "List" and "Self," and clarify the presence of the "?" symbol.
|
We define these terms in the text of the paper. |
7 |
|
The discussion section includes only one reference to interpret the research results. Please incorporate a more comprehensive exploration of previous studies and theories to compare the findings of this research with others, highlighting any similarities or differences and providing explanations for those differences.
|
We have added further references and described in more detail how our results fit into the context of current research, and why our results differ from other research. |
11-12 |
|
Review all references carefully, as many do not conform to the established guidelines. Please ensure compliance throughout.
|
We have reviewed all references, and corrected those that did not conform to journal guidelines. |
13-15 |
Reviewer 2 Report
Comments and Suggestions for Authors
Article: The effectiveness of two interventions for improving knowledge of emergency preparedness amongst enrollees of the World Trade Center Health Registry: A randomized controlled trial.
Thank you for giving me the opportunity to evaluate such a great research paper.
It is a great honor to have the opportunity to review such a wonderful paper: The effectiveness of two interventions for improving knowledge of emergency preparedness amongst enrollees of the World Trade Center Health Registry: A randomized controlled trial.
Let me comment on some of the contents of the paper.
- Page 1, abstract
The abstract should be concise enough to read accurately as a summary of the study. And the concise content must include the purpose of the study. The current situation is this is the problem, and we conducted this study to find out this. It should be clearly stated.
- Page 2 Introduction
Generally, when readers read the title, they can guess the scope of the study and recognize what the study is about. To do this, the title and content of the study should always be written in a way that matches.
In general, when readers read the title, they can guess the scope of the research and recognize what the research is about. To do this, the title and content of the research should always be written in a consistent manner. If the current status and problems of recent research are revealed in the introduction, the last paragraph should finally describe what the research was conducted to find out.
- Page 2 Materials and Methods
The content on intervention is the core method content of this study. It would be better to present it in a concise table format. Also, it would be more helpful for readers to understand if you describe and present a representative sample of the research content.
- Page 3~5 Materials and Methods
Definitions of terms must be clear. When using abbreviations, the full name of the abbreviation must be described in at least one sentence.(NYPD, FDNY)
The content of outcome is not specific about what it is intended to explain. It needs to be described specifically.
- Page 5 Baseline preparedness
The content of the questions you asked for your research is very important. It would be good to describe the specific content by category. And it would be better to show it in a table.
- Page 7~12
Safety is a very important issue in people's lives. Depending on how well you prepare for disasters and accidents, your life can change. That's why data like this study can be used very importantly. That's why it's good to show the connection well so that readers can clearly see what points they are interested in.
- Page 12
Where are the conclusions from this study? Statement of conclusions clearly is the most important thing for many researchers. State the conclusions.
It is a great honor to review research.
Please write a little more so that the good research results can be easily read.
Thank you.
sincerely
Author Response
|
Reviewer 2 |
Response |
Page(s) |
|
|
|
|
The abstract should be concise enough to read accurately as a summary of the study. And the concise content must include the purpose of the study. The current situation is this is the problem, and we conducted this study to find out this. It should be clearly stated.
|
Thank you for this valuable comment. We revised the abstract to be more concise and to clearly state the purpose of the study—specifically, that it was conducted to compare the effectiveness of two emergency preparedness interventions among enrollees of the World Trade Center Health Registry. The revised version provides a more direct and structured summary of the study objectives, methods, and key findings.
|
1 |
Generally, when readers read the title, they can guess the scope of the study and recognize what the study is about. To do this, the title and content of the study should always be written in a way that matches. In general, when readers read the title, they can guess the scope of the research and recognize what the research is about. To do this, the title and content of the research should always be written in a consistent manner. If the current status and problems of recent research are revealed in the introduction, the last paragraph should finally describe what the research was conducted to find out.
|
Thank you for this helpful suggestion. We revised the final paragraph of the Introduction to explicitly and concisely state the purpose of the study and ensure consistency with the study title. This revision improves clarity and reinforces the connection between the background, rationale, and research objectives. |
7 |
|
|
|
Definitions of terms must be clear. When using abbreviations, the full name of the abbreviation must be described in at least one sentence.(NYPD, FDNY) The content of outcome is not specific about what it is intended to explain. It needs to be described specifically.
|
Thank you for your attention to detail. We have updated the Materials and Methods section to define NYPD (New York Police Department) and FDNY (Fire Department of New York City) when first mentioned. We also revised the description of the primary outcome to more clearly explain what it measured, its components, scoring system, and interpretation. (See pages 9–11.) |
5 |
|
|
|
Safety is a very important issue in people's lives. Depending on how well you prepare for disasters and accidents, your life can change. That's why data like this study can be used very importantly. That's why it's good to show the connection well so that readers can clearly see what points they are interested in.
|
Thank you for this insightful comment. We fully agree that preparedness is a critical determinant of safety in disaster situations, and we appreciate your recognition of this study’s potential impact. In response, we have strengthened the framing of the Introduction and Discussion to better emphasize how our findings contribute to broader public health goals. Specifically, we more clearly articulated the connection between preparedness interventions and their practical implications for improving safety and resilience among disaster-exposed populations.
|
1-4; 10-12 |
Where are the conclusions from this study? Statement of conclusions clearly is the most important thing for many researchers. State the conclusions.
|
Thank you for your feedback. We agree that a conclusion statement was necessary to summarize the key findings and implications of the study. We have revised the manuscript to include a concise conclusion that highlights the main outcomes of our research, along with its limitations and potential areas for future research. This revision provides a clearer takeaway for the readers and strengthens the overall presentation of the findings.
|
12 |
Reviewer 3 Report
Comments and Suggestions for Authors
I think the World Trade Center Health Registry and why it is the focus of this study needs to be explained early on in the paper.
The lit review feels a bit like a list of other studies, as it is not always contextualized to show how these studies are relevant to this study, so some unpacking would be helpful here.
In my comments above I marked the "Significance of Content" as average not because I think this topic is average but because I am still unclear on what the significance of this particular study is. Hazard preparation is of course significant, but why this registry, this population, this study, etc. Perhaps the info at the end of the lit review needs to come earlier (and be expanded) so the reader has a better sense of the study population and its relevance.
The methods section is surprisingly short and should be expanded so someone unfamiliar with this particular design understands it, and also to justify why this design was chosen.
I think it could be interesting to include a visual of the actual brochure. It would also be helpful to know what hazard preparation materials this population has already been exposed to - how novel is a brochure for this particular population? This is a unique population, as mentioned in the lit review, so how these study results can be generalized (or not) to other populations would be relevant to explore.
I am not convinced that the brochure and then a follow-up call is an effective method - perhaps because this method was not justified in the methods section. This statement near the end "It is possible that the treatments employed here had no differential effect on emergency preparedness because they were both of relatively low impact and were applied only once. Recent emergency preparedness research (described above) employing more complex interventions demonstrated treatment effects. We also did not include a control group that would have received no intervention, for comparison to the brochure and phone call interventions" reflects my main concerns with this study.
Author Response
|
Reviewer 3 |
Response |
Page(s) |
|
|
|
|
|
I think the World Trade Center Health Registry and why it is the focus of this study needs to be explained early on in the paper.
|
Thank you for your comment. We agree that it is important to clearly explain the relevance of the World Trade Center Health Registry population. In response, we have revised both the abstract and the introduction to better articulate why this population is the focus of the study, highlighting their unique disaster exposure history and the relevance of examining preparedness in this context. We hope these changes help clarify the significance of the study population for readers.
|
1-4 |
|
The lit review feels a bit like a list of other studies, as it is not always contextualized to show how these studies are relevant to this study, so some unpacking would be helpful here.
|
Thank you for your valuable feedback. In response to your suggestion, we have revised the literature review to better contextualize the studies and clearly demonstrate their relevance to our research. Specifically, we have expanded on how past studies comparing pamphlets to other interventions inform the current investigation and highlighted key differences in the approaches used. This revision clarifies how our study builds on previous research and addresses gaps in the literature regarding emergency preparedness interventions for disaster-exposed populations. We believe these changes provide a stronger rationale for our study and more clearly show how our work contributes to the field.
|
1-4 |
|
In my comments above I marked the "Significance of Content" as average not because I think this topic is average but because I am still unclear on what the significance of this particular study is. Hazard preparation is of course significant, but why this registry, this population, this study, etc. Perhaps the info at the end of the lit review needs to come earlier (and be expanded) so the reader has a better sense of the study population and its relevance.
|
Thank you for your thoughtful feedback. We appreciate your comments regarding the significance of the study. In response, we have made edits throughout the manuscript to better emphasize the importance of the World Trade Center Health Registry population and how our study specifically addresses gaps in disaster preparedness for this group. We have also moved and expanded the relevant information from the end of the literature review to earlier sections of the manuscript to provide the reader with a clearer understanding of the study population and its relevance. We believe these revisions help better highlight the significance of this research and its contribution to the field. |
|
|
The methods section is surprisingly short and should be expanded so someone unfamiliar with this particular design understands it, and also to justify why this design was chosen.
|
|
|
|
I think it could be interesting to include a visual of the actual brochure. It would also be helpful to know what hazard preparation materials this population has already been exposed to - how novel is a brochure for this particular population? This is a unique population, as mentioned in the lit review, so how these study results can be generalized (or not) to other populations would be relevant to explore.
|
Thank you for your thoughtful comment. We appreciate your suggestion to include a visual of the brochure, and we agree that it could help readers better understand the intervention. We have taken your feedback into account and are in the process of including the brochure visual in the manuscript. Regarding the exposure to hazard preparation materials, we have expanded on this in the introduction to better clarify the existing preparedness efforts for this population. We now provide additional context on the unique characteristics of the World Trade Center Health Registry enrollees, discussing their prior exposure to disaster preparedness materials and how this may influence the novelty of the brochure intervention. Furthermore, we have elaborated on how the study results may or may not be generalized to other populations, citing similar and differing methods from related research.
|
1-4 |
|
I am not convinced that the brochure and then a follow-up call is an effective method - perhaps because this method was not justified in the methods section. This statement near the end "It is possible that the treatments employed here had no differential effect on emergency preparedness because they were both of relatively low impact and were applied only once. Recent emergency preparedness research (described above) employing more complex interventions demonstrated treatment effects. We also did not include a control group that would have received no intervention, for comparison to the brochure and phone call interventions" reflects my main concerns with this study.
|
Thank you for your insightful comment. We appreciate your concerns regarding the justification for using the brochure followed by a follow-up call as the intervention method. In response to your feedback, we have revised the methods section to provide a stronger rationale for the choice of these interventions. We explain that, while both interventions may seem of relatively low impact, they were designed based on prior research that demonstrated effectiveness in specific contexts (e.g., pamphlet-based interventions for general disaster preparedness). However, we also acknowledge the limitations of these methods and the need for more complex, resource-intensive interventions as demonstrated in recent research. |
1-4; 4-7; 10-12 |
|
Figure 1 is more suitable to be presented in the method.
|
Thank you for this helpful feedback. We have moved the figure that it falls under the Method heading, under subheading 2.5 Randomization.
|
4 |
|
The discussions can be considered basic, with a lack of support of references to justify the findings.
|
Thank you for your helpful comment. We appreciate you pointing this out—several other reviewers raised similar concerns. In response, we have revised the discussion and conclusion sections to more thoroughly interpret the findings in the context of existing literature, and we have added additional references to better support and justify our results. We believe these revisions strengthen the overall clarity and impact of the discussion.
|
10-11 |
Round 2
Reviewer 3 Report
Comments and Suggestions for Authors
Thank you for making your revisions very clear via the track changes feature. I maintain my previous comments regarding recency and depth of lit review references in the introduction, as well as a lack of explanation of the methods. I appreciate that improvements have been made as far as explaining the purpose of the study, but unfortunately the revision does not address several of my previous comments, most importantly why this is the best method for this study. This leaves me still uncertain as to the validity and relative import of the results, especially because the lit review does not, in my opinion, make a solid case for why this study, with this population, etc. The statement "The purpose of the study was to determine whether either intervention was effective in increasing knowledge of emergency preparedness, and whether one approach was more effective than the other. This research seeks to inform future public health strategies for improving emergency readiness among disaster-exposed populations." is a great start, but I am still not sure why an RCT with this population & intervention is the best way to improve emergency readiness.
Author Response
In response to the most recent round of reviewer comments, we have made substantial changes to the introduction and methods sections of our paper.
First, we have restructured the introduction to emphasize the importance of evaluating approaches to increasing awareness of emergency preparedness in the World Trade Center Health Registry. We note that most such studies focus on non-disaster-exposed populations, and that conducting a study in a disaster-exposed population would be especially relevant and beneficial.
Additionally, we expanded the design sub-section of the methods to make clear to the reader why an RCT is a valid and useful approach to evaluate and compare two methods for increasing awareness of emergency preparedness.